# Sources of Stress among Saudi Arabian Nursing Students: A Cross-Sectional Study

**DOI:** 10.3390/ijerph182211958

**Published:** 2021-11-14

**Authors:** Wafaa Aljohani, Maram Banakhar, Loujain Sharif, Fatimah Alsaggaf, Ohood Felemban, Rebecca Wright

**Affiliations:** 1Department of Medical Surgical Nursing, Faculty of Nursing, King Abdulaziz University, Jeddah 21551, Saudi Arabia; wfaljohani@kau.edu.sa; 2Nursing Program, Batterjee Medical College, Jeddah 21442, Saudi Arabia; 3Department of Public Health Nursing, Faculty of Nursing, King Abdulaziz University, Jeddah 21551, Saudi Arabia; ahbbanakher3@kau.edu.sa (M.B.); ofelemban@kau.edu.sa (O.F.); 4Department of Psychiatric and Mental Health Nursing, Faculty of Nursing, King Abdulaziz University, Jeddah 21551, Saudi Arabia; 5Department of Maternity and Child Health Nursing, Faculty of Nursing, King Abdulaziz University, Jeddah 21551, Saudi Arabia; fmalsaggaf@kau.edu.sa; 6School of Nursing, Johns Hopkins University, Baltimore, MD 21205, USA; rebecca.wright@jhu.edu

**Keywords:** nursing students, Saudi Arabia, stress, baccalaureate, bridging

## Abstract

Introduction: Nursing students experience higher levels of stress than those in other health-related disciplines; however, there are limited data exploring stress among these students in a Saudi context. Aim: This study examines sources of stress among nursing students at an academic institution in Jeddah, Saudi Arabia, using a descriptive quantitative cross-sectional research design. Methods: Data were collected from a convenience sample of 500 undergraduate nursing students, with a response rate of 71.8%, using an adapted Stress in Nursing Students (SINS) questionnaire. Results: Nursing student sources of stress fell into three categories: academic concerns, clinical practice, and social factors. Discussion: The results demonstrate commonality between other countries’ sources of stress for nursing students but highlight cultural factors unique to Saudi Arabia. This study shows opportunities for cross-cultural learning and areas needing cultural tailoring to reduce stress among nursing students.

## 1. Introduction

In Saudi Arabia, the nursing profession is advancing, with increasing opportunities to enter training programs, from entry-level diplomas to PhDs. With these advancements come increasing responsibility and changes and refinements of roles, but there is also a challenge to ensure clarity in the scope of practice and sustainability of the nursing workforce [1]. As such, many nursing students may experience a significant level of stress as they train. Though stress among nursing students is a common phenomenon [2], it is imperative that such stressors are fully understood within a cultural context in order to tailor supportive strategies.

Stress is defined as “mental or physical tension or strain” [3]. It is a common issue that can be experienced by anybody, and it can occur when there is a demand for change [3,4,5]. Stress can affect people in most professions. However, nursing students have been found to have the highest levels of stress compared to students of other health professions [6]. High levels of stress can result in negative consequences, such as physical and mental health problems [7]. Nursing students who experience stress may suffer from anger, anxiety, depression, lack of sleep, poor concentration, memory deficiency, and learning difficulties [7,8,9,10,11]. Moreover, stress can negatively affect students’ behaviors, such as their nutritional intake, work productivity, and social interaction [8,9,10,11,12]. Nursing students most commonly experience stress within three domains: academic concerns, clinical practice, and social factors [9,12,13,14].

## 2. Stress Domains

### 2.1. Stress from Academic Concerns

Academic factors comprise course requirements such as excessive content, long theoretical and clinical hours in stressful and changing environments, continual assignments and exams, and fear of failing or underachievement [5,6,8]. In addition, lack of leisure time because of academic load may contribute to stress load [6,9,15]. For example, one study found that 10 to 30 percent of cases of stress resulted from concerns over poor academic performance [3]. As Saudi Arabia continues to develop its nursing programs and opportunities, we must address the potential for high levels of stress to result in negative physical and psychological consequences, which may interfere with nursing students’ academic performance and progress.

### 2.2. Stress from Clinical Practice

The second domain, clinical practice, has been reported as a significant factor in stress among nursing students [16]. Clinical factors include the unfamiliarity of nursing students with the clinical environment, lack of confidence to practice, and fear of making errors while using advanced technology or equipment [8,9,10,11,17]. Other factors identified included clinical area crowdedness and smells and noises experienced by students in a clinical area [18,19]. In addition, nursing students may also witness clients’ suffering or pain, death, and dying, which can lead to high levels of stress [7]. Student–educator relationships in clinical settings have also been reported as a source of stress among nursing students. Common stressors identified included students feeling rejected and unsupported by nurses, negative attitudes, being criticized, and the pressure of proving oneself in the clinical setting [5,20]. In addition to these, inconsistencies between theory and practice and continuous evaluations during clinical rotations were reported [9,11,21].

### 2.3. Stress from Social Factors

The third domain of social factors includes financial and personal issues, for example, the necessity of finding a job to pay for school tuition fees [8,9]. Thus, lack of leisure time is also attributed to the financial difficulties experienced by nursing students [9,15]. Although stress sources among nursing students have been investigated by many studies, most were conducted in western countries, with few addressing stress among nursing students in Saudi Arabia [2,22]. Of those conducted in Saudi Arabia, a recent review [1] examined 19 studies conducted over the last 17 years in relation to the position of the Saudi nursing profession and the recently launched Kingdom 2030 Vision [23].

According to the review, Saudi Arabia faces serious challenges in the form of nursing shortages related to issues in the nursing education system, a poor image of nursing, and an unclear scope of practice. The author suggests that nurse educators should strive to enhance the attractiveness of the nursing profession in order to meet the nation’s significant demand [1]. Addressing potential sources of stress by drawing on the current body of knowledge provides a promising route for investigation. However, in order to provide a high-quality curriculum using problem-based learning that accommodates both the psychological and academic needs of nursing students, we must also examine stress in a Saudi context. Therefore, the aim of the present study is to identify and understand specific stressors within two nursing programs at a Saudi Arabian nursing college as well as highlight where strategies currently used in other contexts may be extrapolated and integrated through culturally appropriate steps.

## 3. Methods

### 3.1. Design

A descriptive, cross-sectional study was conducted. The research protocol was reviewed and approved by the Nursing Research Ethics Committee at the Faculty of Nursing within the academic institution (NREC Serial No: Ref No 1F. 60, April 2017).

### 3.2. Setting and Participants

A convenience sampling strategy was used to obtain a sample according to the inclusion criteria of being a full-time student enrolled in either the baccalaureate (years 2, 3, and 4) or the bridging nursing program (years 1 and 2) at a nursing college of an academic institution in Jeddah, Saudi Arabia. The bridging nursing program at the institution is a 2.5-year internship program designed to further advance diploma-level technical nurses interested in pursuing a baccalaureate degree to become nurses able to work in specialized hospitals, long-term care facilities, and community settings. Nurses who satisfactorily complete the graduation requirements of the bridging program are awarded a BSN degree. This academic educational institution was selected because of its large student population. An estimated sample size of 500 nursing students across baccalaureate and bridging programs was calculated with a 95% confidence interval and a 5% margin of error.

### 3.3. Measurement Tools

This study used the original English language 43-item adapted SINS questionnaire [24]. Three items (30, 37, 41) were removed as they were outside the scope of the present study, leaving a modified SINS questionnaire with 40 items. Item 30 described the relationship of the student with other professionals in clinical training, and items 37 and 41 focused on the student’s finances. Item 30 was removed because the main focus during clinical training for both baccalaureate and bridging level students is their engagement with patients as well as their relationship with the supervising faculty member, and they had a limited relationship with other hospital healthcare professionals due to limited clinical training hours. Item 37 (surviving on a low income) was removed, given the Saudi context, as students do not pay for their degree at governmental academic institutions; additionally, they receive a monthly stipend from the institution during their nursing program years. Item 41 (making less money than friends who are not nurses) does not apply to the Saudi context as most nurses are employed immediately with relatively moderate to high starting salaries.

A pilot study confirmed the internal consistency of the modified questionnaire (Cronbach’s alpha of 0.922). The modified SINS questionnaire has two parts. The first part collects demographic data (4 items), while the second part includes 40 items rated on a Likert scale of 1 (Not at all stressful) to 5 (Extremely stressful) (see supplemental attachment). The five sub-scales are as follows: (1) Relationships with patients and their families (8 items); (2) Intellectual challenge (11 items); (3) Personal and social concerns (9 items); (4) Clinical challenge (5 items); (5) Relationships with superiors and colleagues (7 items). The 5 dimensions of SINS have been previously reported in the original scale and in previously conducted studies [24]. According to the different subscales investigated, for each, a raw score and a scaled score were calculated. Raw scores correspond to the sum of items’ scores, while the scaled scores (min = 1; max = 5) correspond to raw scores divided by the number of items.

### 3.4. Data Collection

The researchers approached the students with paper-based questionnaires at the target college during students’ free time. The research team explained the purpose of the study and answered any questions. This included outlining how and why the participants were involved, how long the questionnaire would take to complete, and the intended use of the results. The participants were assured of the confidentiality of their data and that participation was completely voluntary. Following this, informed consent forms were distributed to participants to sign. Questionnaire completion took approximately 15–20 min. Data collection took place over three months from April to June 2017, and completed questionnaires and copies of consent forms were stored in a secured room in a locked cabinet to which only the research team had access.

### 3.5. Data Analysis

Data were analyzed using SPSS (Statistical Package for Social Sciences) version 25(IBM Corp, Armonk, NY, USA). Descriptive analyses were used to calculate percentages, means, frequencies, and standard deviations. Independent *t*-tests were used to measure differences in SINS between baccalaureate and bridging level and demographic variables. A *p*-value < 0.05 was taken as significant at 95% confidence intervals (95% CI).

## 4. Results

### 4.1. Characteristics

In total, 359 questionnaires were completed and retrieved (a response rate of 71.8%). Of the total sample, 84.4% were aged less than 25 years, while 11.4% were aged between 26 and 30 years, and only 4.2% were between 31 and 35 years. All the study participants were of the female gender. Regarding marital status, 90.5% were not married, while 8.4% were. The sample consisted mostly (78.5%) of baccalaureate nursing students, and only 21.5% were bridging students; 82.2% had less than one year of clinical experience, whereas 10.3% reported that they had clinical experience ranging between 5 and 9 years. The sample characteristics are summarized in Table 1.

### 4.2. Levels of Stress Factors and Demographic Characteristics

There was a negative relationship between age and stress levels related to intellectual challenge. The stress related to this factor was reduced with age. The highest levels of stress were found among students aged equal to or less than 25 years (3.16 ± 0.62), followed by 26–30 years (2.99 ± 0.74) and 31–35 years (2.78 ± 0.85). The difference was statistically significant (*p* = 0.028). Furthermore, statistically high scores related to intellectual challenge were found among non-married students (*p* = 0.018). The results also revealed that the stress related to clinical challenge was greatest among third-year undergraduate students and lowest among first-year bridging students (*p* = 0.005; see Table 2).

The incidence of stress for both baccalaureate and bridging students, according to the SINS tool, was found to be statistically significant (*p* = 0.043) (Table 3). However, in the analysis of the five sub-scales, three were found to be statistically significant stress factors for both student groups: intellectual challenge (*p* = 0.012), personal and social concerns (*p* = 0.035), and clinical challenge (*p* = 0.024).

### 4.3. Differences in the Levels of Stress within Baccalaureate and Bridging Programs

The analysis identified differences in levels of stress between the two nursing programs (Table 4). In 8 of the 40 SINS items, baccalaureate students scored higher than bridging students, with statistically significant differences. Significant differences were found in examinations and placement grading (*p* = 0.001), fear of making a mistake in clinical placements (*p* = 0.001), fear of poor job prospects (*p* = 0.035), lack of free time (*p* = 0.017), not having enough money for entertainment (*p* = 0.011), fear of failing the course (*p* = 0.003), lacking time for entertainment (*p* = 0.003), and feeling responsible for what happens to patients (*p* = 0.041).

## 5. Discussion

Using a modified SINS questionnaire [24], this study aims to identify the difficulties leading to the development of stress experienced by nursing students when they enter either the baccalaureate or bridging program at a nursing college in Saudi Arabia. While we found some shared and high-level stressors in both programs (intellectual and clinical challenges and personal and social concerns), there were some unique differences in the stress levels found between bridging and baccalaureate students that may be explained by age difference and experience.

Baccalaureate nursing students had higher levels of stress than bridging students with regard to lack of free time, fear of making a mistake in clinical placements, fear of failing the course, having no time for entertainment, examinations and placement grading, feeling responsible for what happens to patients, not having enough money for entertainment, and fear of poor job prospects. Bridging students are often older and may experience less stress because they are more mature than younger students; they are, therefore, more likely to have developed problem-solving skills that can help them adjust to the demands of their studies [16,25]. Furthermore, most of the bridging students have had previous clinical experiences. This is borne out by the finding that fear of failing the course, examinations, and placement grading were reported as the highest stress factors for baccalaureate students.

The results reveal that, overall, stress related to intellectual challenge was reduced with age. In addition, non-married students experienced lower levels of stress than married students. It is possible that the security of a family could decrease stress, as we noted a correlation between younger age and marriage, whereby most students aged less than 25 years were baccalaureate students and single. A similar finding was reported in a descriptive correlational study conducted in Jordan by [26], which showed that younger students had higher levels of stress than senior students.

When considering the findings within the three domains underpinning this study (academic, clinical, and social stressors), one “academic stressor” explanation for intellectual and clinical challenges, scoring high among baccalaureate students, could be related to exposure to a new experience, where there is a heavy academic and clinical workload [26]. However, bridging students’ high scores in these two items can be explained by the fact that these students study under an accelerated program that is more intensive than the baccalaureate program [16].

The wider literature also speaks of the stress of examinations and grading among nursing students, with one explanation being the large amount of academic knowledge required in a short space of time to pass exams [2,27]. Examinations, in general, were a key area of stress. For example, [27] found that more than 80% of nursing students considered lengthy exams a cause of moderate to extreme stress, while [28] found that students considered many of their examinations ambiguous. One solution offered by [29] is for academics to modify the assessment methods used in their courses and consider implementing student-friendly approaches, such as reducing the frequency of exams to reduce overall intellectual-challenge-related stress.

The second domain of interest is clinical practice, which has been highlighted by many researchers [9,17,26,30]. In this study, baccalaureate students reported that fear of being responsible for what happens to patients and fear of making a mistake in clinical placements were factors contributing to high stress levels. This was also reported by two studies [2,16] that found that baccalaureate students consider fear of causing harm to a patient to be a source of stress. Within this and other studies, nursing students have shown self-awareness about their lack of knowledge of and familiarity with highly technical settings, medical terminologies, and the procedures performed in clinical settings; though such insights are important for safety, they create stress [5,26]. Since, in this study, both bridging and baccalaureate students experienced stress derived from clinical training, it is worth emphasizing the importance of endowing students with the skills required for caring for patients in clinical practice. This could include simulations in the lab prior to clinical exposure, which may reduce students’ stress regarding clinical practice [31].

In addition, the present study found that third-year baccalaureate nursing students experienced the highest levels of stress related to clinical challenge, while first-year baccalaureate nursing students experienced the lowest. A study suggested that stress is linked to the course as they found that students in clinical training courses had the highest levels of stress [26]. In our study, third-year nursing students’ high scores in the area of intellectual challenge could be related to them studying advanced and heavy courses for which clinical training is required, such as adult medical surgical nursing, pediatric nursing, and maternity nursing courses. The study also argued that students’ stress can be related to the knowledge and skills expected by the faculty for the students to act as professional nurses as they approach graduation [26].

The third domain within which we considered our findings was personal and social concerns, which scored the lowest among the three stress domains. This domain was also identified as an area of stress induction in a study in Saudi Arabia, which noted relationships with parents, personal issues, and health issues of a family member as particularly pertinent [2]. In another study [15], conducted in Egypt, changing living environment was a personal factor of stress for most participants. Similarly, in the present study, many of the baccalaureate and bridging students had moved away from home to attend the nursing college in Jeddah. Although absence or distance from supportive family units is a widely recognized source of stress [9] in the Saudi context, the resulting feelings of stress are likely exacerbated due to the cultural focus, traditions, and networks around a community, particularly for women [29].

Stress derived from balancing the requirements of work and life was found in both this study and throughout the literature [2,15], with lack of free time being seen as more stressful than lacking time for entertainment. There is an obvious overlap between this domain and those of academic and clinical stressors. For example, long hours of study, exams, and assignments may negatively affect students’ free and leisure time [8,9]. In one study, [32] it was found that nursing students in Norway spend an average of 30 h per week studying. Thus, the skills of nursing students should extend beyond academic and clinical domains and into time management, with the added benefit of potentially reducing academic stress and increasing students’ life satisfaction and performance [15].

### 5.1. Nursing Implications from a Global Perspective

The SINS tool is considered an effective reporting measurement based on its extensive global use and potential to extrapolate the local Saudi findings to a wider context. Previous studies using the SINS tool with student nurse stress have taken place in Australia [33], China [34], Malaysia [35], Pakistan [36], Singapore [37], Hong Kong [24], Spain [38], and the United Kingdom (UK) [39]. Though the reporting of these studies varies (for example, not all provide total scores), they provide insights into our own setting, along with a lens for the comparison of stress across international nurse training programs.

The findings of the present study show that the greatest area of stress was clinical challenges (e.g., fear of making a mistake in clinical placement). This was also found to be the case in Malaysia [35] and China [34], whereas in Singapore, clinical and financial factors were found to be most stressful [37], and in the United Kingdom and Spain, financial stress was higher than clinical stress [39]. This may speak to differences in access to the program, the availability of scholarships, and the stage of the development of the training programs or the supportive measures implemented. For example, the UK study was published in 2003, when nursing programs were fully funded—scores may, therefore, differ in light of changes to bursary payments made in 2017 [40].

To understand these clinical challenges further, the study by [41] in Pakistan gives some context, finding that 84% of their nursing student participants considered “unclear career future” to be a source of stress. In Saudi Arabia, most students who graduate from a nursing program work long hours, ranging from eight to twelve hours per shift, including both night and day shifts, which may constitute a unique cultural factor in stress about the future. While 8 and 12 h work shift patterns are common in many other countries, according to labor law in Saudi Arabia, the standard working hours are 8 h daily and 48 h per week, with a reduction in work hours during Ramadan to six hours a day and 36 h a week for Muslim workers [42].

Female Saudi nurses find working hours and night shifts particularly difficult, as working hours here are longer than in other professions more culturally conventional for Saudi women, such as teachers, who work no more than 6 h per day without night duty [43]. This becomes a significant contributor to stress as the majority of Saudi nurses are young mothers, and this may make a career in nursing a less-appealing career choice [44]. This is particularly pertinent, as nursing is viewed as a questionable profession among some members of the Saudi public due to the unfavorable mixed-gender workplace environment, exacerbated by long hours, night shifts, and weekend duties [44]. This can result in stigmatization, suspicious views, and even disrepute aimed at female nurses, who are further challenged by the belittling of their profession and the perception that nurses are not independent practitioners and instead work in a position subservient to physicians [45].

Included within the “academic domain” are the high levels of academic stress that must be understood within a cultural and global context. An indirect indication of the high SINS academic stress score and a unique source of stress facing both Saudi baccalaureate and bridging students is the requirement that all healthcare programs in Saudi Arabia be taught in the English language. Healthcare students are taught in English in order to prepare them for work in different healthcare organizations that use the English language [46]. This source of stress coheres with the findings of other Arab-context studies, such as a Jordan-based investigation into the perceived learning challenges of the nursing and health sciences (NHS), which found that students struggle with English language proficiency in terms of both general and medical terminology [47].

Similarly, a study looking at the difficulties faced by first-year medical students in a Saudi University found that poor English language skills was ranked as the second-highest difficulty faced by the students [48]. Language requirements can impact student stress to such an extent that one study found it to be a contributing factor in student dropout rates [49]. Thus, we may be able to understand the present study’s findings on stress derived from fear of harming patients and unfamiliar environments as exacerbated by the bilingual work context. This is a prime example of the limited generalizability of stress mitigation strategies employed in other cultural contexts into a Saudi one. Given the increased potential for life-threatening errors based solely on miscommunication, research into overcoming language barriers may prove useful [50].

In the context of the findings of this study, future research should explore the roles within nursing institutions that are culturally focused and the facilitation of counseling and academic advice for students who encounter different sources of stress during their programs. Interventions addressing the three domains, with scope to cater to the different life stages, ages, and cultural challenges associated with different programs, are needed as this study highlights that sources of stress within the same domain may be experienced in vastly different contexts.

In addition, research on this topic has predominantly captured female perspectives, and the proliferation of men in nursing suggests important data may be derived from cross-sectional studies assessing sources of stress among male nursing students as well as comparisons between male and female experiences. Additionally, it is worth replicating the study to assess sources of stress in both baccalaureate and bridging nursing programs at different private and governmental institutions across the Kingdom of Saudi Arabia. This will help determine whether our findings, understood as “cultural”, arise elsewhere and whether strategies can be generalized nationally.

### 5.2. Limitations

As noted, this study addressed a female sample, as male nursing programs have only recently been established and the courses are not mixed. This may limit the generalizability of the results to other Saudi nursing students who may face different sources of stressors; the inclusion of more males opens a new avenue for exploration, as noted above. The use of a cross-sectional research design may have limited the inferences that could be made in terms of causal effects between sample variables. Hence, the use of future longitudinal studies with larger, mixed-gender sample sizes is likely to increase the understanding of how Saudi students experience stress. Finally, the assessment of stress among students was based on a self-reported questionnaire, which is subject to social desirability bias.

## 6. Conclusions

This study identified higher levels of stress among baccalaureate nursing students compared to bridging students, with sources of stress most commonly related to fear of making mistakes in clinical practice, intellectual challenge, examinations, and grading as well as the lack of free time. Cultural challenges may underpin the extent and experience of stressors, and, therefore, the provision of culturally tailored academic and psychological support in colleges and other tertiary educational institutions is highly recommended in order to reduce students’ stress levels. Moreover, educational institutions should consider the impact of their teaching and learning environment and consider implementing active teaching and learning activities that target the highlighted areas of stress.

## Figures and Tables

**Table 1 ijerph-18-11958-t001:** Participants’ characteristics.

Parameter	Category	Frequency	Percentage
Age (years)	≤25	303	84.4
	26–30	41	11.4
	31–35	15	4.2
Gender	Female	359	100
	Male	0	0
Marital status	Married	30	8.4
	Not married	325	90.5
	Divorced	4	1.1
Level of education	Baccalaureate	301	83.8
	Bridging	57	15.9
	Missing	1	0.3
Year	1st bridging year	41	11.4
	2nd bridging year	15	4.2
	2nd year	86	24.0
	3rd year	53	14.8
	4th year	65	18.1
	Missing	99	27.6
Clinical experience	<1 year	295	82.2
	1–4 years	23	6.4
	5–9 years	37	10.3
	10–14 years	3	0.8
	Missing	1	0.3

**Table 2 ijerph-18-11958-t002:** Factors of stress among nursing students.

Factor	Category	Overall Stress (SINS Score)			Relation with Patients			Intellectual Challenge			Personal and Social Concerns			Clinical Challenge			Relationship with Superiors		
		M.	SD	*p*-Value	M.	SD	*p*-Value	M.	SD	*p*-Value	M.	SD	*p*-Value	M.	SD	*p*-Value	M.	SD	*p*-Value
Age	≤25	119.55	22.06	0.118	2.90	0.73	0.486	3.16	0.62	0.028 *	3.05	0.69	0.115	3.08	0.69	0.060	2.68	0.73	0.661
	26–30	114.61	28.31		2.83	0.88		2.99	0.74		2.82	0.76		2.93	0.83		2.72	0.87	
	31–35	109.00	29.45		2.68	0.85		2.78	0.85		2.90	0.86		2.68	0.79		2.51	0.94	
Marital status	Married	117.00	28.01	0.153	2.90	0.89	0.642	2.98	0.70	0.018 *	2.95	0.78	0.290	3.03	0.90	0.450	2.75	0.81	0.410
	Not married	118.96	22.78		2.88	0.74		3.15	0.64		3.03	0.71		3.05	0.70		2.68	0.75	
	Divorced	96.75	13.28		2.53	0.48		2.32	0.37		2.50	0.47		2.60	0.63		2.21	0.79	
Year	1st bridging	109.51	30.95	0.089	2.70	0.93	0.267	2.84	0.83	0.064	2.72	0.81	0.233	2.79	0.82	0.005 *	2.61	0.90	0.421
	2nd bridging	121.07	18.31		2.99	0.67		3.13	0.54		3.12	0.64		3.03	0.82		2.79	0.86	
	2nd year	119.07	21.64		2.99	0.75		3.10	0.60		3.00	0.64		3.08	0.72		2.67	0.67	
	3rd year	123.43	25.22		3.03	0.83		3.23	0.68		2.96	0.71		3.37	0.72		2.89	0.80	
	4th year	117.78	22.40		2.89	0.68		3.05	0.54		3.00	0.78		3.06	0.60		2.69	0.78	
Clinical experience	<1 year	119.69	22.19	0.222	2.91	0.73	0.337	3.16	0.62	0.103	3.05	0.70	0.119	3.09	0.70	0.083	2.69	0.73	0.966
	1–4 years	115.13	29.21		2.72	0.88		3.06	0.73		2.92	0.80		2.96	0.96		2.66	0.93	
	5–9 years	112.57	27.51		2.83	0.86		2.93	0.76		2.78	0.71		2.84	0.69		2.63	0.86	
	10–14 years	107.00	11.79		2.29	0.62		2.61	0.69		3.33	0.78		2.40	0.80		2.57	1.08	

* Statistically significant result (*p* < 0.05).

**Table 3 ijerph-18-11958-t003:** Levels of stress among baccalaureate versus bridging nursing students (independent *t*-test).

Scale	Total			Undergraduate		Bridging		*p*-Value
	Range	Mean	SD	Mean	SD	Mean	SD	
SINS total score	51, 200	118.25	24.29	119.64	22.10	112.84	28.24	0.043 *
Relation with patients and their families	1.00, 5.00	2.93	0.78	2.90	0.73	2.79	0.87	0.289
Intellectual challenge	1.27, 5.00	3.07	0.65	3.16	0.62	2.93	0.77	0.012 *
Personal and social concerns	1.33, 5.00	2.95	0.72	3.05	0.70	2.84	0.78	0.035 *
Clinical challenge	1.40, 5.00	3.09	0.73	3.08	0.70	2.85	0.82	0.024 *
Relationship with superiors and colleagues	1.00, 5.00	2.72	0.78	2.68	0.74	2.66	0.88	0.831

* Statistically significant result (*p* = 0.05).

**Table 4 ijerph-18-11958-t004:** Levels of stress by SINS questionnaire item among baccalaureate versus bridging nursing students (independent *t*-test).

Item	Baccalaureate		Bridging		*p*-Value
	Mean	SD	Mean	SD	
(7) Examinations and placement grading	3.47	1.22	2.82	1.20	<0.001 *
(9) Fear of making a mistake in clinical placements	3.69	1.22	2.81	1.33	<0.001 *
(16) Fear of poor job prospects	3.03	1.27	2.65	1.13	0.035 *
(22) Lack of free time	3.78	1.22	3.35	1.29	0.017 *
(28) Not having enough money for entertainment	3.08	1.37	2.58	1.29	0.011 *
(32) Fear of failing the course	3.63	1.30	3.07	1.27	0.003 *
(34) Having no time for entertainment	3.47	1.29	2.91	1.30	0.003 *
(37) Feeling responsible for what happens to patients	3.44	1.32	3.05	1.27	0.041 *

* Statistically significant result (*p* = 0.05); significant differences were found in questions 7, 9, 16, 22, 28, 32, 34 and 37.

## Data Availability

The data presented in this study are available on request from the corresponding author.

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
