# Peer review of "Sources of Stress among Saudi Arabian Nursing Students: A Cross-Sectional Study"

_ijerph, 2021, doi:10.3390/ijerph182211958_

Round 1
Reviewer 1 Report
Congratulations on the study. It is certainly methodologically appropriate and it is appreciated that they have stated the limitations so clearly. However, they need to present their results with more clarity due to the significant gender bias and add in the discussion aspects related to the reconciliation of work and personal life and its influence on the stress of the students.
I especially want to congratulate section 5.1, since it gives the manuscript the international character that such a local study can have.
Author Response
Response 1: All participants were of female gender. This point has been added to the text and table 1 on page 4 and highlighted in yellow. Furthermore, with regards to “ the discussion aspects related to the reconciliation of work and personal life and its influence on the stress of the students” this point is addressed in more details on page 8 under section 5.1. Nursing Implications from a Global Perspective, paragraph 4:
“Female Saudi nurses find working hours and night shifts particularly difficult, as working hours here are longer than in other professions more culturally conventional for Saudi women, such as teachers, who work no more than 6 hours per day without night duty [43] . This becomes a significant contributor to stress as the majority of Saudi nurses are young mothers, and this may make a career in nursing a less appealing career choice [44] . This is particularly pertinent, as nursing is viewed as a questionable profession among some members of the Saudi public due to the unfavorable mixed-gender workplace environment, exacerbated by long hours, night shifts and weekend duties [44]. This can result in stigmatization, suspicious views and even disrepute aimed at female nurses, who are further challenged by the belittling of their profession and the perception that nurses are not independent practitioners and instead work in a position subservient to physicians [45] .”

Reviewer 2 Report
I found this paper enjoyable to read. It is very well written and structured. In the attached PDF version of your draft I have made comments in 'Track Change' relating to some minor questions I had. I feel that the information obtained from this study could go on to inform the design of stress-reduction and wellbeing interventions in your nursing programmes. I encourage you to consider doing this.

Author Response
Point 1: Saudi nursing profession and the recently launched Kingdom 2030 Vision (Reference required).
Response 1: Reference added as number 23 within the text and in the reference list highlighted in yellow.
Point 2: Commented [U1]: Does the Ethics Committee approval reference number need to be given here?
Response 2: Ethics Committee approval reference number was added on page 3 and highlighted in yellow.
Point 3: Commented [U2]: Were these 5 subscales established and named as part of the current study or have they been previously reported? It would be useful to clarify that point.
Response 3: The 5 dimensions of SINS have been previously reported in the original scale and in the previous conducted studies (Waston et al., 2010), was added on page 3 and highlighted in yellow.
Point 4: Commented [U3]: Does this need a reference?
Response 4: No
Point 5: Commented [U4]: Were all of the participants female? Although you discuss gender in the limitations paragraph (5.2), I feel it would be useful to state that here
Response 5:
All participants were of female gender. This point has been added to the text and table 1 on page 4 and highlighted in yellow.
Point 6: Commented [U5]: I couldn’t see the information but I wondered about any correlation between older baccalaureate students and bridging students – i.e. were there more similarities in subscale scores? Including that data would support your interpretations on this point. However, your numbers may be much too small to assist useful analysis.
Response 6:
Thank you for the suggestion however, as you mentioned our numbers were too small to assess this analysis perhaps it could be considered in future studies.

Reviewer 3 Report
General comments:
Thank you for opportunity for reviewing this interesting paper. Research partially adheres to STROBE guidelines. I believe that the topic of the manuscript is very interesting.
I believe that this manuscript doesn´t qualify for acceptance at this time and should be improved for publication in IJERPH.
Specific comments:
- Writing
The writing, structure and organization of the manuscript is in accordance with the guidelines.
- Title
The title reflects the content and problem studied.
- Abstract
The abstract reflects reflects the manuscript and provide an informative and balanced summary of what was done and what was found.
The kind of study should be in methods
- Key Words
The keywords are representative of the subject studied and exposed. Bridging aren´t Mesh Terms
- Background
The background reflects the state of the art in relation to the study. The objective of the study is mentioned, as well as the justification for the choice and importance of studying this theme.
- Methods
There is detailed description of the research methods used. The design is correct and it is possible to validate the veracity of the results
They describe the setting, location but don´t describe relevant dates or including periods of recruitment.
Eligibility criteria, sources and methods of selection of participants are explained.
the explanation of statistical methods should be improved!! normality is calculated? that justifies the use of non-parametric measures?
sample size is explained
Authors don´t describe any efforts to address potential sources of bias
- Findings
The results shown are concrete and detailed, explaining how to obtain this information and what scientific evidence it has.
The number of persons in each phase of the study is reported
Authors should consider using a flow chart.
Statistical analysis needs to be improved
- I think that author should be calculated intra-group differences in table 2.
- It is important for the interpretation of the results and further analysis.
- Discussion
The key results of the discussion are concrete. In addition, it includes the main strengths and weaknesses in relation to other studies, discussing important differences in the results.
Limitations have been exposed
It´s clear and concise. the conclusions are in line with the objective
- References
The references used are correct, the vast majority dating back less than ten years.
Author Response
Kindly refer to the attached word document.
